

**Biogeochemical climatology for the Southern Benguela Upwelling System,**
**constructed from *in situ* monitoring data**
Stephanie de Villiers
Centre for Coastal Paleoscience, Nelson Mandela University, Port Elizabeth, South Africa
*Correspondence to*: Stephanie de Villiers (steph.devilliers@gmail.com)



**Abstract.** An annual and a seasonal biogeochemical climatology had been constructed for the
Southern Benguela Upwelling System, from *in situ* data collected along a 12 station
monitoring line, sampled at monthly intervals from 2001 to 2012. The monitoring line
reaches a maximum offshore distance of almost 190 km, with monitoring station depths
ranging from 27 to 1 465 m. In addition to temperature, salinity and oxygen CTD profile
data, archived monitoring data for the macro-nutrients (phosphate, nitrate + nitrite, silicate)
and chlorophyll-a was evaluated. The climatologies exhibit clear spatial and seasonal
variability patterns for all parameters, that yield important insight into the SBUS upwelling
cycle. These data sets comprise valuable additions to our knowledge base, and will aid both
future modelling efforts and studies of biogeochemical processes in upwelling systems. Data
for the constructed climatologies has been made available via the PANGAEA Data Archiving
and Publication database at http://doi.pangaea.de/10.1594/PANGAEA.882218.
**Keywords:** Upwelling; Benguela; climatology; biogeochemistry; oxygen; nutrients;
chlorophyll-a.



## 1 Introduction

The world's four Eastern Boundary Upwelling Systems (EBUSs), the California, Humboldt, Canary/Iberian and the Benguela, are regions of intense biological production and ocean-atmosphere exchange of $CO_2$. Together they account for more than 10% of new oceanic primary production and provide more than 20% of the world's commercial fish catches (Chavez and Toggweiler, 1995; Fennel, 1999; Pauly and Christensen, 1995). In addition, they serve as important sources of nutrients to adjacent oligotrophic subtropical gyres. The strength of the biological pump in these large coastal upwelling systems depends on the availability of nutrients in the photic zone. Nutrient supply and export, in turn, are controlled by complex and climate-sensitive physical forcing factors, such as thermal stratification, coastal wind fields and upwelling intensity (Bakun, 1990; McGowan et al., 1998; Wang et al., 2005; Barross et al., 2014; Bakun et al., 2015; Rykaczewski et al., 2015). Increased stratification (e.g. in response to warmer surface ocean temperatures) will result in decreased nutrient transport into the photic zone, whereas enhanced upwelling intensity (e.g. in response to increased atmospheric temperature gradients) will increase nutrient transport into the photic zone. The latter may also potentially increase offshore transport of nutrients.

In order to understand variability in the productivity of EBUSs, and to predict the potential impact of climate and global change, the relationship between nutrient availability and coastal upwelling processes need to be much better understood than it is at present (Cury and Roy, 1989; Bakun et al., 2010; 2015; Botsford et al., 2006). Coupled physical-biogeochemical models have played a vital role in advancing our understanding of these systems (Skogen, 1999; Qian, 2012; Gutknecht et al., 2013). Such modelling efforts benefit greatly from the availability of climatological data. Data for parameters such as surface ocean temperature and chlorophyll-a levels are relatively easily measured and monitored with remote sensing techniques (Fiúza et al., 1982; Chen et al., 2012; Tim et al., 2015). Construction of comparative climatologies for key biogeochemical parameters such as dissolved oxygen and nutrients, however, require systematic *in situ* sampling. The scarcity of data sets that are appropriate for the construction of biogeochemical climatologies, is problematic. This data gap limits our understanding of upwelling system dynamics (Palacio et al., 2004; Garcia-Reyes e al 2015).





The Benguela Upwelling System (BUS) is located along the south-west coast of
Africa (Figure 1). It consists of two parts that differ in their mean seasonality, atmospheric
drivers and large-scale climate modes (Agenbag and Shannon, 1988; Tim et al., 2015), the
Northern Benguela Upwelling Systems (NBUS) and the SBUS. At least eight discrete
upwelling cells have been identified within the BUS (Figure 1; Lutjeharms and Meeuwis,
1987). The Lüderitz cell, in the NBUS, is the major upwelling cell and displays the highest
frequency of occurrence and strength (Lutjeharm and Meeuwis, 1987; Gutknecht et al.,
2013). The NBUS and the Lüderitz cell have been the subject of numerous, but irregular,
biogeochemical research cruises and studies (Dittmar and Birkicht, 2001; Emeis et al., 2004;
Kuypers et al., 2005; van der Plas et al., 2007; Mohrholz et al., 2008; Sohm et al., 2011;
Noble et al., 2012; Nagel et al., 2013; Flohr et al., 2014;). The SBUS had been monitored for
longer and in a more systematic manner than the NBUS, including along the St. Helena Bay
Monitoring Line (SHBML) across the Columbine upwelling cell (Figure 1). The monitoring
data for the SBUS, however, had been available to a limited extent (Hutchings et al., 2009;
Lamont et al., 2015) and most of the bottle data had not been quality controlled or compiled
into a distinct long-term monitoring data set. The latter issue had recently been addressed.
This biogeochemical monitoring data have now been used to construct both an annual and a
seasonal climatology for the SBUS, accompanied by analogous climatologies for
temperature, salinity and oxygen, from CTD data.
**2 Data processing**
**2.1 Sampling location, frequency and data availability**
The location, bottom depth and sampling depths for the 12 monitoring stations along the
SHBML are given in Table 1 and illustrated in Figure 1. The sampling frequency was
approximately monthly, with the actual cruise occurrences summarized in Table 2. The
months of March and September were sampled every year from 2001 to 2012. The other
months were sampled at least 9 times during the 12 year monitoring period, with the
exception of November, which was sampled only 4 times. The bottle sampling depths used
to construct a climatology for dissolved phosphate ($PO_4^{3-}$ or P), nitrate + nitrite ($NO_3^- + NO_2^-$,
or TN), silicate ($SiO_2$ or Si) and chlorophyll-a (Chl-a) are listed in Table 1. These depths
were sampled routinely; depths that were sparsely sampled were not included in the
construction of the climatology, to avoid possible bias. CTD data (T, S and $O_2$) was
condensed to 1 m sampling intervals, for the surface-to-bottom depth ranges indicated in





Table 1. The following data was not used in the construction of the climatologies: (i) CTD
data from 2009 to 2012, because of concerns about infrequent sensor calibrations since 2009,
(ii) Bottle data for 2001 to 2003, because of concerns about nutrient data quality prior to
2004. The total number of discrete data points used to construct a climatology for each of the
parameters are given in Table 3.
CTD data and bottle samples were collected according to published protocols and analytical
methods (Lamont et al., 2015; Ismail et al., 2015). CTD measurements were carried out
using three different vessels and multiple Sea-Bird Electronics SBE 911 systems according to
international standards, including Winkler titrations to calibrate dissolved oxygen profiles
(Lamont et al., 2015). Bottle samples for nutrient analysis were collected using the same
protocols, and analysed with the same equipment and analytical methods (Astoria Analyzer
Series 300; Ismail et al., 2015), over the monitoring period. It is difficult to definitively
establish long-term data uncertainty and accuracy over such an extended period of time, for
any of the parameters, not least of all because of intrinsic spatial and temporal variability.
The most practical method with which to evaluate data uncertainty, for the purpose of
constructing this climatology, is crossover analysis of measurements made at the furthest
offshore station (Station 12), at depth (1 350 m), where temporal and spatial variability is
expected to be small. Since temporal variability cannot be ruled out, this provides an upper
estimate of the uncertainty associated with analytical methodologies. The signal-to-noise
ratios (calculated as the average/standard deviation, using all measurements at this depth,
over the monitoring period) are as follows: ~ 65 for T, ~ 1634 for S, ~ 24 for $O_2$, ~ 6.5 for P,
~ 15.6 for TN and 17.3 for Si. These values suggest that long-term analytical data
uncertainty is appropriate for the construction of a climatology.
**2.2 Data reduction and construction of seasonal and annual climatologies**
The monthly monitoring data was first reduced to a seasonal climatology for each parameter
(X), by calculating average monthly values for each parameter at each of the 12 stations, at
each of the depths provided in Table 1 (results illustrated in Figure 2 and 3). The seasonal
climatologies were then used to construct annual climatologies for each parameter (results
illustrated in Figure 4, 5 and 6), as described below.





At each of the 12 monitoring stations, every parameter is represented by a set of calculated
average monthly values, $X_M[S_i, D_Z]$, at each of the discrete sampling depths listed in Table 1:
X =    T = Temperature (°C), S = Salinity (psu), $O_2$ = Dissolved oxygen (ml $O_2$ $dm^{-3}$), P =

5          Dissolved phosphate ($\mu$mol P $dm^{-3}$), TN = Dissolved nitrate + nitrite ($\mu$mol N $dm^{-3}$),

6          Si = Dissolved silicate ($\mu$mol Si $dm^{-3}$), or Chl = Chlorophyll-a ($\mu$g $dm^{-3}$)

$S_i$ =    Station number ($S_1$ to $S_{12}$)
$Y_j$ =    Year ($Y_{first}$ to $Y_{last}$; 2001 to 2008 for CTD data, 2004 to 2013 for bottle data)
M =    Month (1 to 12, or January to December)
$D_Z$=    Depth (5 m etc. as listed in Table 1, 1 m increments for CTD data)
For Station $S_i$ at discrete sampling depth $D_l$, the seasonal climatology for parameter X
consists of the following 12 monthly average values (for January to December):
$X_{JAN}[S_i, D_Z]$    $X_{FEB}[S_i, D_Z]$    $X_{MAR}[S_i, D_Z]$    $X_{APR}[S_i, D_Z]$    $X_{MAY}[S_i, D_Z]$    $X_{JUN}[S_i, D_Z]$
$X_{JUL}[S_i, D_Z]$    $X_{AUG}[S_i, D_Z]$    $X_{SEP}[S_i, D_Z]$    $X_{OCT}[S_i, D_Z]$    $X_{NOV}[S_i, D_Z]$    $X_{DEC}[S_i, D_Z]$
with    $X_M[S_i, D_Z]$    = AVERAGE$\{X_M[S_i, Y_{first}, D_Z]$........$X_M[S_i, Y_{last}, D_Z]\}$
The seasonal climatology is then reduced to an annual climatology for each parameter
($X_{ANN}$), by calculating the average of the twelve monthly $X_M[S_i, D_Z]$ values (at discrete
sampling depths, for each of the 12 stations):

$X_{ANN}[S_i,D_Z]$    = AVERAGE$\{X_{JAN}[S_i, D_Z]; X_{FEB}[S_i, D_Z].....X_{DEC}[S_i, D_Z]\}$

The standard deviation ($X_{sd}[S_i,D_Z]$) associated with the calculated annual averages represents
the intra-annual variability in each parameter, i.e. the magnitude and location of variability:

$X_{sd}[S_i,D_Z]$    = STANDARD DEVIATION$\{X_{JAN}[S_i, D_Z]; X_{FEB}[S_i, D_Z].....X_{DEC}[S_i, D_Z]\}$

For each parameter X the annual climatology, as a function of depth along the monitoring
transect, is represented by a matrix of values:
$X_{ANN}[S_1, 5]$    $X_{ANN}[S_1, 10]$    $X_{ANN}[S_1, 23]$
$X_{ANN}[S_2, 5]$    $X_{ANN}[S_2, 10]$    $X_{ANN}[S_2, 21]$    $X_{ANN}[S_2, 30]$
$X_{ANN}[S_3, 5]$    $X_{ANN}[S_3, 10]$    $X_{ANN}[S_3, 20]$    $X_{ANN}[S_3, 30]$    $X_{ANN}[S_3, 55]$    $X_{ANN}[S_3, 70]$
$X_{ANN}[S_4, 5]$    $X_{ANN}[S_4, 10]$    $X_{ANN}[S_4, 20]$    $X_{ANN}[S_4, 30]$    $X_{ANN}[S_4, 55]$    $X_{ANN}[S_4, 79]$



etc.
The calculated seasonal climatologies for each of the parameters are available as a text file on
the PANGAEA database (http://doi.pangaea.de/10.1594/PANGAEA.882218*).*

## 3 Description of the climatologies

### 3.1 Annual climatologies

The annual average and intra-annual variability patterns (i.e. standard deviation) for T, S, $O_2$
and the dissolved nutrients are illustrated with cross shelf vertical transects, for the upper 600
m, in Figure 4 and 5. The data for chlorophyll-a is plotted to a 40 m depth only (Figure 6). At
the surface (5 m depth), temperature ranges from 13.1 to 18.2°C along the 190 km long
transect from Station 1 (inshore) to Station 12 (offshore), and salinity from 34.77 to 35.40
(Figure 4). Along the bottom, temperature ranges from 11.3 to 3.0°C, and salinity from 34.77
to 34.65, in an offshore direction. Surface water nutrient concentrations vary from high
inshore to much lower levels offshore: 1.55 to 0.36 µmol dm$^{-3}$ for P, 10.6 to 1.4 µmol dm$^{-3}$
for TN and 16.2 to 2.6 µmol dm$^{-3}$ for Si (Figure 4). Surface water chlorophyll-a ranges from
values > 8 µg dm$^{-3}$ inshore to < 1 µg dm$^{-3}$ offshore (Figure 6). High chlorophyll-a values (> 3
µg dm$^{-3}$) are also restricted to the upper ~ 25 m depth interval at the inshore stations. The
highest nutrient values, and lowest oxygen levels, are observed in the bottom waters
underlying Stations 2 to 5 (Figure 4). The calculated annual climatology for Stations 1 to 5
yield bottom water ranges of 2.0 to 2.5 µmol dm$^{-3}$ for P, 20 to 25 µmol dm$^{-3}$ for TN, and 28 to
33 µmol dm$^{-3}$ for Si. Calculated values for the annual dissolved oxygen climatology in these
bottom waters fall within the 1 to 2 ml dm$^{-3}$ range.
Water is known to upwell from a depth of 200 to 300 m in the BUS (Nelson and
Hutchings, 1983). The 10°C isotherm shallows from a depth of 293 m at Station 12, to 46 m
at Station 4 (Figure 4). If this isotherm is used to trace the flow path of upwelling water, it
suggests the following "pre-formed" values (for 10°C water): salinity ~ 34.8, $O_2$ ~ 4.1 ml
dm$^{-3}$, P ~ 1.3 µmol dm$^{-3}$, TN ~ 16 µmol dm$^{-3}$ and Si ~ 12 µmol dm$^{-3}$. At Station 1, the
shallowest station, the average nutrient content is 1.68 µmol dm$^{-3}$ for P, 13.1 µmol dm$^{-3}$ for
TN and 19.0 µmol dm$^{-3}$ for Si. This is much higher than the values for warm offshore surface
water (P = 0.40 µmol dm$^{-3}$, TN = 1.80 µmol dm$^{-3}$ and Si = 2.88 µmol dm$^{-3}$ for the upper 30 m





at Station 12). The average nutrient content of water at Station 1 is also higher, for P and Si, than the "pre-formed" values at a depth of 293 m at Station 12 detailed above. This indicates enrichment of shallow coastal water in P and Si relative to upwelled water, from either land-based sources or from entrainment of the nutrient-enriched bottom water that is present at Stations 1 to 6.

There are considerable spatial differences in intra-annual variability for all the parameters, expressed as the standard deviation associated with calculated annual average values (Figure 5 and 6). However, physical and biogeochemical parameters exhibit distinctly different patterns. Intra-annual variability in temperature and salinity is highest within the upper 100 m of the water column at the offshore stations, while oxygen and the nutrients are more variable at the inshore stations and at sub-surface to bottom water depths. Chlorophyll-a is most variable at shallow depths close to the coast (Figure 6). A detailed evaluation of these zones of high variability, and discussion of the factors responsible, is not the focus of this data discussion. Suffice it to say that areas of high intra-annual variability in temperature and salinity are most likely associated with seasonal changes in the depth of the thermocline and intrusion of mesoscale features, such as warm, salty Agulhas rings. High variability in the biogeochemical parameters are most likely the result of the seasonal cycle of organic matter production and regeneration.

## 3.2 Seasonal climatologies

The January to December seasonal climatologies exhibit temporal and spatial variability, for all of the parameters, that are typical of coastal upwelling systems (Figure 2, 3 and 8). The seasonal upwelling cycle is most pronounced at Stations 2, 3 and 4 (Figure 7). The $10^{\circ}$C isotherm in the temperature climatology again serves as a useful indicator of the evolution of the seasonal upwelling cycle. At Stations 4 to 12 the $10^{\circ}$C isotherm is present throughout the water column all year round (Figure 2). At Station 4, the $10^{\circ}$C isotherm reaches its maximum depth ($\sim$ 60 m) mid-winter (July), starts to shoal in August and reaches it shallowest depth ($\sim$ 30 m) mid-summer (January) (Figure 7). At Station 3, the $10^{\circ}$C isotherm makes it appearance in the bottom waters in August, in September at Station 2, and in October at Station 1. This suggests the timing and evolution of the seasonal upwelling cycle. The deeper $9^{\circ}$C isotherm starts to shoal towards shallower depths even earlier, during the months of May and June, according to the seasonal climatology at Station 4 (Figure 7).



Another pronounced seasonal change evident in the physical water mass parameters, is the
movement of an upper ocean warm and salty (S > 34.9) water mass, that migrates closer to
shore during the winter months and that is located further offshore in summer (Figure 2).
The mixing of warm, oxygen-rich and nutrient-depleted surface water to deeper depth
during winter is a prominent feature in the seasonal climatologies at Stations 2 to 4 (Figure
7). This is followed by the upwelling of nutrient-rich, oxygen-depleted, bottom water, from
September onwards. Surface water chlorophyll-a values start to increase with the spring
upwelling cycle and is reduced during the winter months (Figure 8). The depth of the
chlorophyll-a rich surface layer reaches its deepest depth in early summer, November to
December. This period of peak summer productivity evident in the chorophyll-a climatology
is followed by a further reduction of oxygen, and enrichment of nutrients, in the bottom
waters at the inshore stations in later summer/early autumn (Figure 7), resulting from the
respiration of organic matter.
**4 Conclusions**
Annual and seasonal climatologies constructed from long-term monitoring data exhibit well-
resolved spatial and temporal changes. This demonstrates the importance of long-term
monitoring efforts, and illustrates what is achievable with systematic sampling and
measurement strategies, even for biogeochemical parameters such as dissolved nutrients.
This data set can be used to optimize future and ongoing monitoring efforts. It contains a
wealth of information for the study of biogeochemical cycles and processes in this and
analogous upwelling systems, and it should prove invaluable to coupled physical-
biogeochemical modelling efforts.










**Data availability**
Data for the constructed climatologies have been made available via the PANGAEA Data Archiving and
Publication database at http://doi.pangaea.de/10.1594/PANGAEA.882218.
**Author contribution**
The author identified the value of the unused archived data sets used in this study, retrieved the data, performed
all data rescue tasks such as quality control and data reduction, and conceived of and wrote this manuscript.
**Acknowledgements**
Numerous present and past employees of the Department of Environmental Affairs and Marine and Coastal
Management were involved in sample collection, analysis and data archiving; most notable have been the long-
term efforts of Christien Illert, Gavin Tutt and Marcel van der Bergh.

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



Figure 1:  The Benguela Upwelling System (shaded green area), off the west coast of southern Africa.  The
upwelling cells identified by Lutjeharms and Meeuwis (1987) are indicated by the stippled yellow lines.  The St
Helena Bay Monitoring Line (SHBML) is indicated by the line of green circles, representing the 12 monitoring
stations; the shelf bathymetry along this line and the sample bottle depths used to construct climatologies for
nutrients and chlorophyll-a, are shown in the inset.
Figure 2: Monthly climatologies for temperature, salinity and oxygen.
Figure 3: Monthly climatologies for P, N and Si.
Figure 4:  Cross-shelf transects for the annual climatology of the parameters temperature, salinity, dissolved
oxygen and the three dissolved nutrients, constructed as outlined in the text.  Data is plotted for the upper 600 m
only, to accentuate the upper ocean gradients.
Figure 5:  Cross-shelf transects for intra-annual variability, or the standard deviation associated with the annual
averages for temperature, salinity, dissolved oxygen and the three dissolved nutrients.
Figure 6:  Annual climatology and intra-annual variability, expressed as the standard deviation, for chlorophyll-
a along the SHBML, shown for the upper 40 m of the water column only.
Figure 7: Seasonal changes, as a function of depth, in temperature, dissolved oxygen, phosphate and nitrate +
nitrite, for Stations 2, 3 and 4, constructed as discussed in the text.
Figure 8: Average chlorophyll-a profiles along the SHBML, illustrated for each month and the upper 50 m of
the water column



**Table 1**: Monitoring station locations, bottom depth, standard bottle sampling depths and CTD depth ranges

used to construct the climatology

| Station | LAT (°S) | LONG (°E) | Bottom (m) | Offshore distance (km) | Standard bottle depths (m) | CTD depth range (m) |
|---|---|---|---|---|---|---|
| 1 | 32.300 | 18.311 | 27 | 3 | 5 - 10 - 23 | 5 - 23 |
| 2 | 32.310 | 18.273 | 32 | 7 | 5 - 10 - 21 - 30 | 5 - 30 |
| 3 | 32.332 | 18.178 | 77 | 16 | 5 - 10 - 20 - 30 - 50 - 70 | 5 - 70 |
| 4 | 32.374 | 17.991 | 107 | 35 | 5 - 10 - 20 - 30 - 50 - 79 | 5 - 100 |
| 5 | 32.416 | 17.809 | 152 | 53 | 5 - 10 - 20 - 30 - 50 - 73 - 100 | 5 - 145 |
| 6 | 32.464 | 17.611 | 192 | 73 | 5 - 10 - 20 - 30 - 50 - 75 - 100 - 150 - 180 | 5 - 180 |
| 7 | 32.502 | 17.422 | 244 | 92 | 5 - 10 - 20 - 30 - 50 - 100 - 235 | 5 - 235 |
| 8 | 32.573 | 17.194 | 285 | 115 | 5 - 10 - 20 - 30 - 50 - 100 - 200 - 275 | 5 - 275 |
| 9 | 32.618 | 16.991 | 310 | 134 | 5 - 10 - 20 - 30 - 50 - 100 - 200 - 305 | 5 - 305 |
| 10 | 32.661 | 16.804 | 392 | 153 | 5 - 10 - 20 - 30 - 100 - 370 | 5 - 370 |
| 11 | 32.706 | 16.622 | 560 | 172 | 5 - 10 - 20 - 30 - 100 - 540 | 5 - 540 |
| 12 | 32.745 | 16.434 | 1 465 | 190 | 5 - 10 - 20 - 30 - 100 - 1 350 | 5 - 1 350 |

26





2 **Table 2** SHBML monitoring cruises occurrences, from 2001 to 2012

| YEAR | JAN | FEB | MAR | APR | MAY | JUN | JUL | AUG | SEP | OCT | NOV | DEC |
|------|-----|-----|-----|-----|-----|-----|-----|-----|-----|-----|-----|-----|
| **2001** | yes | yes | yes | yes | yes | yes | yes | yes | yes | yes | no | yes |
| **2002** | yes | no | yes | yes | yes | yes | no | no | yes | yes | yes | yes |
| **2003** | yes | yes | yes | no | yes | yes | yes | yes | yes | yes | no | yes |
| **2004** | yes | yes | yes | yes | yes | yes | yes | yes | yes | no | yes | yes |
| **2005** | yes | yes | yes | yes | yes | yes | yes | yes | yes | yes | no | yes |
| **2006** | yes | yes | yes | yes | yes | yes | no | yes | yes | yes | no | yes |
| **2007** | yes | yes | yes | yes | yes | no | yes | yes | yes | yes | no | yes |
| **2008** | yes | yes | yes | no | yes | yes | no | yes | yes | yes | yes | no |
| **2009** | yes | yes | yes | yes | yes | yes | yes | yes | yes | yes | no | yes |
| **2010** | yes | yes | yes | yes | no | yes | yes | yes | yes | yes | no | no |
| **2011** | no | yes | yes | no | yes | yes | yes | yes | yes | no | no | yes |
| **2012** | yes | no | yes | yes | yes | yes | yes | yes | yes | yes | yes | no |

26

27





1  **Table 3** Number of discrete measurements used to construct a climatology for each of the parameters (as

2  discussed in the text, data from 2004 to 2012 for nutrients and chl-a; and CTD data from 2001 to 2008).

| Station | $PO_4^{3-}$ | $NO_3^- + NO_2^-$ | $SiO_2$ | Chl-a | CTD (T, S and $O_2$) |
|---|---|---|---|---|---|
| 1 | 231 | 231 | 231 | 216 | 1 536 |
| 2 | 293 | 293 | 293 | 261 | 2 016 |
| 3 | 406 | 409 | 409 | 287 | 6 336 |
| 4 | 413 | 414 | 414 | 279 | 9 216 |
| 5 | 460 | 460 | 460 | 276 | 13 536 |
| 6 | 483 | 483 | 483 | 284 | 16 896 |
| 7 | 509 | 509 | 509 | 282 | 22 176 |
| 8 | 519 | 519 | 519 | 279 | 26 016 |
| 9 | 516 | 513 | 517 | 276 | 28 896 |
| 10 | 546 | 537 | 550 | 274 | 35 136 |
| 11 | 540 | 535 | 540 | 231 | 51 456 |
| 12 | 543 | 539 | 543 | 237 | 129 216 |
| **TOTAL** | **5 459** | **5 442** | **5 468** | **3 182** | **342 432** |

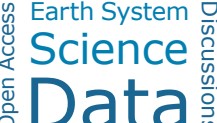

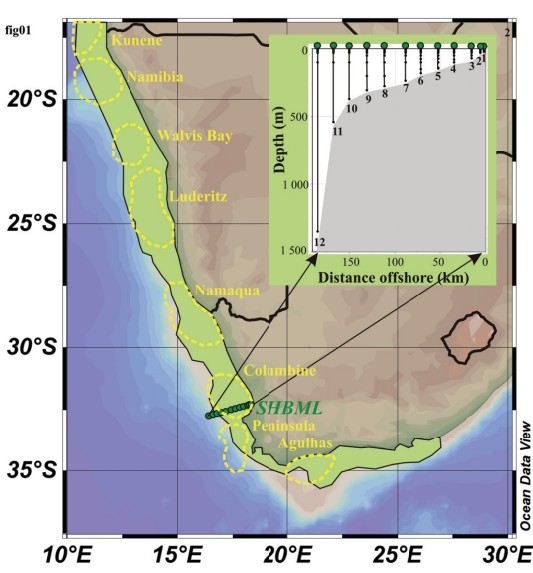









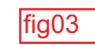

fig04







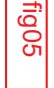





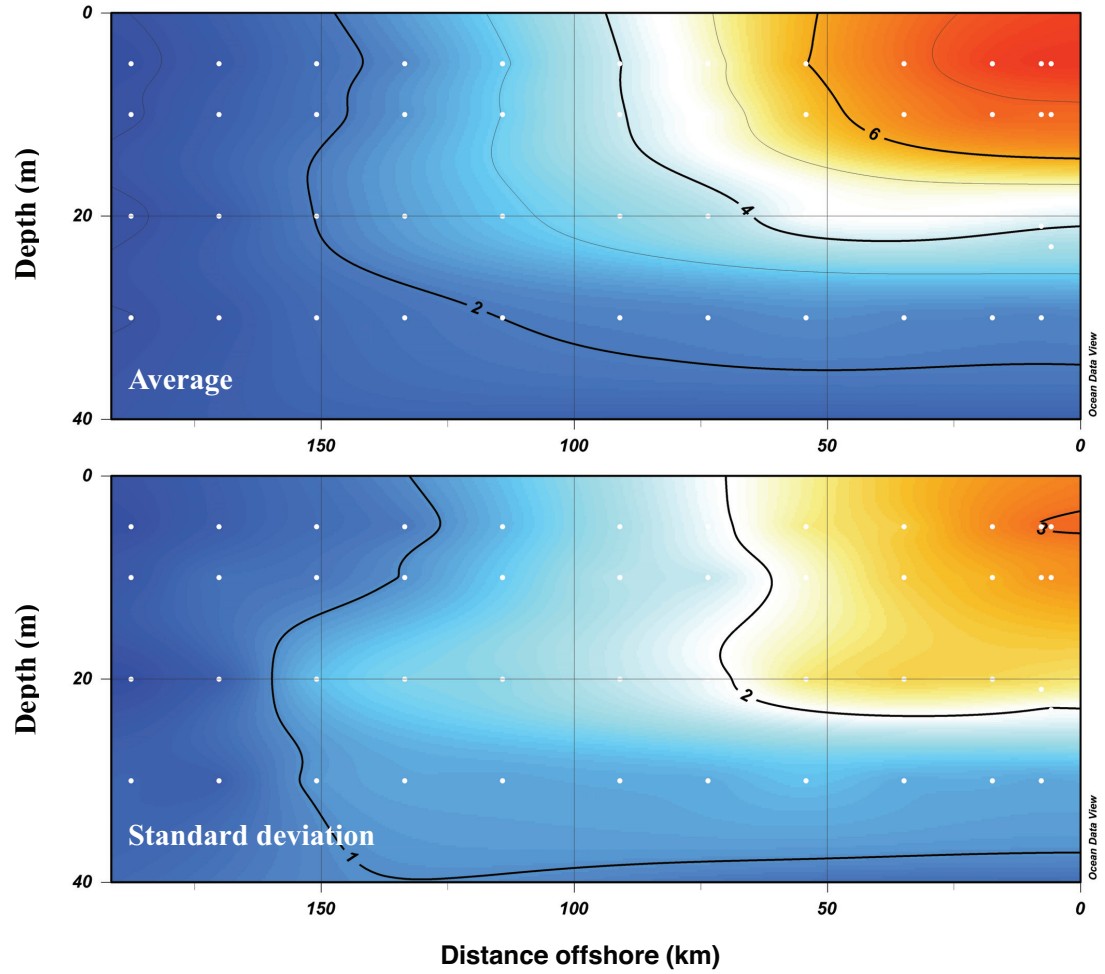



fig07





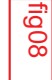

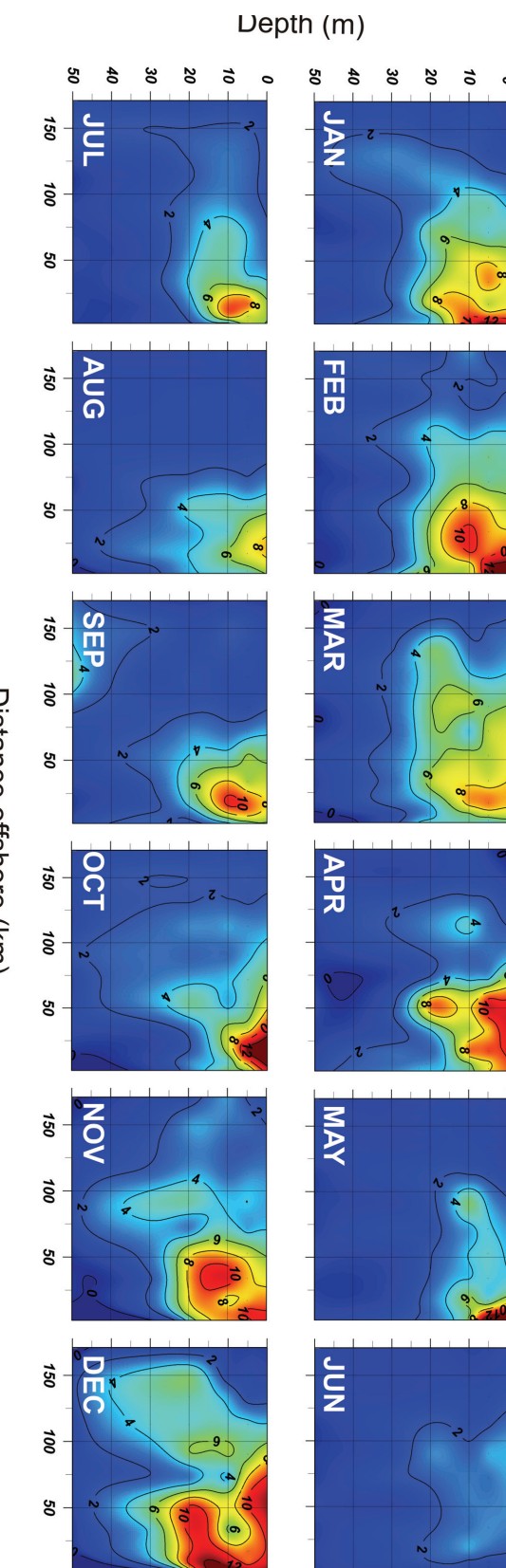