# Peer review of "Biogeochemical climatology for the Southern Benguela Upwelling System,"

_Earth System Science Data, 2018_

## Referee Comment (RC1) · Anonymous Referee #1 · 6 Jun 2018

This paper discusses a dataset obtained from a monitoring line over a 12-year period within the southern Benguela upwelling zone. The data have been discussed previously by Hutchings et al. (2009), Lamont et al. (2015) and Ismail et al. (2015), but the data set is important because it is the first long-term, consistent monitoring line from this region since the 1960s, when the South African Sea Fisheries Research Institute carried out regular seasonal surveys of the St Helena Bay region. Since then, there have been only limited, short-term, time-series surveys of particular regions in the Benguela. The early work was described in Shannon (1985) and Chapman and Shannon (1985), and since then additional time-series studies have been carried out by Bailey and Chapman (1991) and Monteiro and van der Plas (2006). These studies

are not mentioned in the paper, which concentrates only on monthly and annual averaged climatologies. While these are generally useful, I am not particularly convinced of their usefulness in this region, which is known to be strongly affected by short-term variability in the local wind field (see e.g., Taunton-Clark, 1979; Johnson and Nelson, 1999), and where inter annual variability is also important.

Given that this is a paper about the data set, I was annoyed that the author did not give details of the methodology used for either analyses or data calibration. This is a major omission. It is not enough to say that details are given in other papers. I want to know how you did the data reduction, what criteria were used to remove suspect data, how many comparisons of CTD oxygen and salinity samples were done, whether the nutrient samples were frozen or analyzed on board ship, and something about the ranges and standard deviations of the data. Given the 12-year time period, it may be that not all of this information is available, and the author does say this, but I certainly want to know as much about the sampling and data management as possible. A list of signal-to-noise ratios for the deepest sample taken along the line doesn't help if you don't give the actual concentrations. Note also that there is a strong movement towards giving nutrient and oxygen data in $\mu$moles/kg, rather than $\mu$moles/dm-3.

As a reviewer, I was asked to check the data quality. Having registered with PANGAEA especially for this purpose, I logged in and was told I am not allowed access to the data set, which rather negates the purpose of this review. So my rating of the data quality is colored by this (I wanted to leave this blank but the system would not allow me to do so). However, from the available metadata, there are some discrepancies between the paper and the database. For example, while the paper discusses data from 12 stations, the database apparently contains data from only 10 (the two deepest stations are missing), similarly, the station positions in PANGAEA differ slightly from those listed in Table 1, although generally only in the third decimal place.

I'm sure that the data are important, and they should be announced and made available, but if nobody is going to be able to access them what is the point?

Minor comments: 1. P3, lines 20-21 – is this really true for this part of the Benguela? The Sea Fisheries Research Institute (now part of the South African Department of Environmental Affairs) has carried out an enormous amount of research into the hydrography of the St Helena Bay region in particular since the 1950s. 2. P4, line 4 – it would help to put the sites on Fig. 1. 3. Data are plural! 4. While it is easy to make continuously shaded color figures that look pretty, in this case I think it is unnecessary and detracts from the usefulness of the figures. The contour lines are what one needs to see, and because the same color scheme is used throughout the paper, color changes and contour lines do not generally correspond, and no color scales are given. It was almost impossible to see contours in the deep blue regions of the figures, for example any temperatures <10°C.

References:

Bailey, G.W. and P. Chapman (1991). Short-term variability during an anchor station study in the southern Benguela upwelling system: chemical and physical oceanography. Prog. Oceanogr., 28, 9-37.

Chapman, P., and L. V. Shannon (1985), The Benguela ecosystem, Part 2. Chemistry and related processes, Oceanogr. Mar. Biol. Ann. Rev.,23, 183–251.

Hutchings L, Van der Lingen CD, Shannon LJ, Crawford RJ, Verheye HMS, Bartholomae CH, van der Plas AK, et al. The Benguala Current: an ecosystem of four components. Prog. Oceanogr. 83: 15-32 (2009).

Ismail HE, Agenbag JJ, de Villiers S, Ximba BJ. Relation between upwelling intensity and the variability of physical and chemical parameters in the Southern Benguel Upwelling System. Int. J. Ocean. Dx.doi.org/10.1155/2015/510713 (2015).

Johnson, A. and G. Nelson (1999). Ekman estimates of upwelling at Cape Columbine based on measurements of longshore wind from a 35 year tme-series. South Afr. J. Mar. Sci. 21, 433-436.

Lamont T, Hutchings L, van den Berg MA, Goschen WS, Barlow RG. Hydrographic variability in the St Helena Bay region of the southern Benguela ecosystem. J. Geophys. Res: Oceans 120: 2920-2944, doi:10.1002/2014JC010619 (2015).

Monteiro, P. M. S., and A. K. van der Plas (2006), Low Oxygen Water (LOW) variability in the Benguela system: Key processes and forcing scales relevant to forecasting, in Benguela: Predicting a Large Marine Ecosystem, Large Mar. Ecosyst. Ser., vol. 14, edited by V. Shannon et al., pp. 71–90, Elsevier, Amsterdam).

Shannon, L. V. (1985), The Benguela ecosystem, Part 1. Evolution of the Benguela, physical features and processes, Oceanogr. Mar. Biol. Ann. Rev., 23, 105–182.

Taunton-Clark, J. (1979). The formation, growth and decay of upwelling tongues in response to the mesoscale wind field during summer. In: South African Ocean Colour and Upwelling Experiment, ed L.V. Shannon, pp 47-61, Sea Fisheries Research Institute, Cape Town.
* * *

---

## Editor Comment (EC1) · D. J. Carlson (Editor) · 16 Jul 2018

After discussion with the author about lack of suitable data access, I believe the author has decided, appropriately in this case, to withdraw the paper.

---

## Author Comment (AC1) · 8 Aug 2018

The paper discusses a biogeochemical climatology dataset that has not been discussed and published previously (as erroneously stated by the reviewer). Some of the original data used to construct the climatology had by discussed by Hutchings et al (2009), Lamont et al (2015) and Ismail et al (2015) – however, the climatology discussed in this paper is unique, new and different from the data discussed in these papers. The reviewer apparently did not realize the importance difference between a climatology and the primary datasets used to construct it. The ESSDD paper has been downloaded almost a 100 times and viewed 350 times since appearing 2 months ago,

and I have received almost 20 requests for data access, all of which suggest that this climatology is considered very useful for this region, by quite a number of scientists, contrary to the views of the reviewer.

The paper is about a constructed climatology, not a presentation of primary datasets; detailed analytical methodologies pertaining to the primary datasets are described in the papers referenced. In my opinion, that is sufficient and also ensures that those papers and their authors will be credited accordingly. The paper does outline the data reduction processes followed, in detail. The latter includes calculation of and presentation of standard deviations, in graphical format as well as in the data text file.

Almost 20 people approached me for data access, as described on the PANGAEA website for data that is embargoed until publication, none of the individuals had similar complaints to the reviewer and all of them were comfortable identifying their identity.

I have approached PANGAEA in regards to updating the datafile, to include data for the two stations discussed in the paper, but that were not included in the original PANGAEA datafile. Data for all 12 stations will therefore be freely available to anybody - who consider this a useful dataset - upon publication.

The reviewer was free and welcome to contact me for data access, as described on the PANGAEA website. Unfortunately the reviewer preferred to remain anonymous. In contrast, the author is sharing data and attempting to publish it on a free open access platform. If people are only willing to voice their opinions about such efforts if they are allowed to remain anonymous, and also want to access such datasets on the condition that they be allowed to remain anonymous, then what is the point of open access, indeed ? How will such attitudes encourage data sharing and constructive scientific discussion? I informed the Editor that I would rather withdraw the manuscript than be forced to share data with anonymous reviewers prior to publication, since then the paper has been downloaded almost 100 times and the data has been widely shared. Clearly, the dataset is considered of value, by many people.
The items listed in the "Minor comments" section can easily be attended to if the editor decides that this paper is publishable. My personal preference is for the "pretty" shaded color figures to remain as is, although I may be tempted to change the color shading to pink.